



# Assimilation of chlorophyll data into a stochastic ensemble simulation for the North Atlantic ocean

Yeray Santana-Falcón[1], Pierre Brasseur[1], Jean Michel Brankart[1], and Florent Garnier[2]

[1]Université Grenoble Alpes, CNRS, IRD, Grenoble INP, IGE, (Grenoble), France
[2]LEGOS, University of Toulouse, CNRS, IRD, CNES, UPS, (Toulouse), France

**Correspondence:** Yeray Santana-Falcón (yeray.santana-falcon@univ-grenoble-alpes.fr)

**Abstract.**

Satellite-derived surface chlorophyll data are daily assimilated into a three-dimensional 24 member ensemble configuration of an online-coupled NEMO-PISCES model for the North Atlantic ocean. A one-year multivariate assimilation experiment is performed to evaluate the impacts on analyses and forecast ensembles. Our results demonstrate that the integration of data improves surface analysis and forecast chlorophyll representation in a major part of the model domain, where the assimilated simulation outperforms the probabilistic skills of a non-assimilated analogous simulation. However, improvements are dependent on the reliability of the prior free ensemble. A regional diagnosis shows that surface chlorophyll is overestimated in the northern limit of the subtropical North Atlantic, where the prior ensemble spread does not cover the observation's variability. There, the system cannot deal with corrections that alter the equilibrium between the observed and unobserved state variables producing instabilities that propagate into the forecast. To alleviate these inconsistencies, a one-month sensitivity experiment in which the assimilation process is only applied to model fluctuations is performed. Results suggest the use of this methodology may decrease the effect of corrections on the correlations between state vectors. Overall, the experiments presented here evidence the need of refining the description of model's uncertainties according to the biogeochemical characteristics of each oceanic region.

## 1 Introduction

Estimating the biogeochemical state of the ocean has become fundamental under the current climate change context due to its key role mediating global carbon stocks (e.g., Houghton et al., 2001). Currently, the optimal combination of observational data with the dynamical equations embedded in models through data assimilation (DA) is the most comprehensive strategy to meet this goal. Therefore, there is a growing effort towards the development of effective DA techniques to improve hindcasts, forecasts, nowcasts, and scenario simulations of ocean biogeochemistry (e.g., Brasseur et al., 2009; Yu et al., 2018; Fennel et al., 2019). At present, the Copernicus Marine Environment Monitoring Service (CMEMS) delivers DA biogeochemical products for only selected regions (von Schuckmann et al., 2019), though the operational production of the data-assimilated biogeochemical state of the ocean is one of its challenging core objectives.





In order to achieve model / data integration it is of utmost importance to explicitly identify the structure of the uncertainties

that affect the model and the observations (Lahoz et al., 2010). In this sense, ensemble methods (e.g., Bessières et al., 2017) are designed to provide a statistical description of the inaccuracies associated with a complex model system by describing the evolution of the probability density function (PDF). An appropriate approach to perform ensemble simulations is by introducing stochastic noise into the (deterministic) model equations to simulate the effect of the uncertainties. Stochastic parameterizations have been used in meteorological forecasting (e.g., Buizza et al., 1999; Leutbecher et al., 2017), and are becoming the standard

procedure for climate modelling (see Palmer, 2012; Berner et al., 2017). In oceanography, the implementation of this type of probabilistic approach has increased in the last decade (e.g., Brankart et al., 2015; Juricke et al., 2017), although its application in physical-biogeochemical models is quite unusual.

In a precursory study, stochastic perturbations were applied into a deterministic solution of the Mercator Ocean (http://www.mercator-ocean.fr.) North Atlantic 1/4° configuration of the NEMO-PISCES coupled model to parameterize selected

model uncertainties associated with some poorly-resolved processes (see Garnier et al., 2016, for more details). Ensemble simulations involving 60 members were performed using a probabilistic version of the NEMO-PISCES simulation for the year 2005. An objective diagnosis of this ensemble simulation showed its probability distribution is quite consistent with ocean color observations in the most productive regions of the North Atlantic, a prerequisite to undertake DA applications.

Ocean color data have been successfully used in DA procedures for improving the simulation of nutrients and primary

production in ocean models (e.g., Gregg, 2008; Ciavatta et al., 2011; Ford et al., 2012; Fontana et al., 2013; Teruzzi et al., 2018). However, none of the latter studies explicitly incorporates the uncertainties in the ocean biogeochemistry introduced by stochastic approaches. In this context, the overarching aim of the present work is to investigate to what extend the parameterizations developed in Garnier et al. (2016) can be implemented to build a complete 4D assimilation system using ocean color data that will update the state-of-the-art of biogeochemical DA. Our strategy will rely on the daily integration of surface

chlorophyll (Chl-*a* hereafter) data within the latter probabilistic solution. For that end, 24 trajectories of the original ensemble are daily updated by a square root algorithm based on the SEEK (singular evolutive extended Kalman) filter (Pham et al., 1998; Brasseur and Verron, 2006) using daily composites of ocean colour observations extracted from MERIS (MEdium Resolution Imaging Spectrometer). Following this strategy, a one year experiment is performed in order to investigate the effects of the assimilation in contrasted periods (e.g., bloom vs nutrient-depleted periods) throughout the annual cycle.

The paper is structured as follows: section 2 presents the coupled model, the assimilation scheme, and the validation metrics. Section 3 presents the results of the experiment, and provides a probabilistic assessment as compared it with a non-assimilated ensemble simulation. A discussion of the most relevant outputs is carried out in section 4. In particular, we assess how the DA system based on parameterized uncertainties can reduce the model uncertainties, and evaluate its performance at selected regions. Lastly, a summary, conclusions and future perspectives are proposed in section 5 in which we suggest directions for

next possible developments.





## 2 Material and methods

### 2.1 Hydrodynamical model

The assimilation system presented here is based on a realistic three-dimensional physical-biogeochemical simulation. The physical component is simulated using the primitive equation free-surface ocean circulation model NEMO (Nucleus for European Modeling of the Ocean, version 3.4; Barnier et al., 2006; Madec et al., 2015), whose prognostic variables are temperature, salinity and the three-dimensional velocity fields.

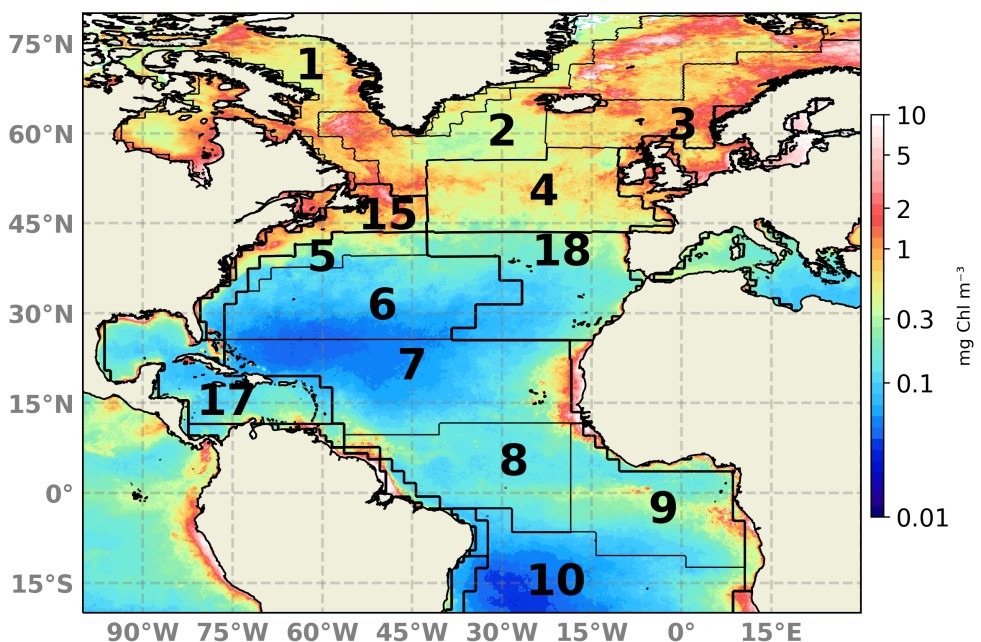

**Figure 1.** Schematic map of the North Atlantic basin showing the *NATL-025/PISCES* domain. Longhurst et al. (1995) biogeochemical provinces are indicated and numbered. A 2018 yearly composite of sea surface Chl-*a* is superimposed.

The model configuration is a duplicate of the North Atlantic configuration developed within the framework of the project DRAKKAR (referred to here as NATL025; Barnier et al., 2006, https://www.drakkar-ocean.eu), which covers the North Atlantic region from 20° S to 80° N and 98° W to 23° E (*Fig.* 1). The numerical grid has a horizontal resolution of a quarter of degree, and 46 geopotential levels in the vertical from surface to 6000 m depth. Such an eddy-resolving resolution enables some rough representation of mesoscales features, which are key elements for primary production (Oschlies and Garçon, 1998; Lévy et al., 2012). The dynamical component is forced by ERA-INTERIM atmospheric fields (Brodeau et al., 2010). This configuration has already been coupled to biogeochemistry modules and evaluated in recent numerical studies (e.g., Ourmières et al., 2009; Doron et al., 2011, 2013; Fontana et al., 2013; Garnier et al., 2016).





## 2.2 Biogeochemical model

The biogeochemical component coupled to hydrodynamics is PISCES (Pelagic Interaction Scheme of Carbon and Ecosystem Studies, version 2; Aumont et al., 2015). PISCES is a complex carbon-based model that simulates marine biological productivity and carbon biomass based upon five main nutrients: nitrate, ammonium, phosphate, silicate and iron. Its architecture includes 24 biogeochemical variables grouped into four main compartments: nutrients, phytoplankton, zooplankton, and detritus. PISCES has been used in global simulations (e.g., Bopp et al., 2015), environmental studies (e.g., Brasseur et al., 2009), climate studies (Lefort et al., 2015), basin scale studies (e.g., Jose et al., 2014) and, more recently, in regional scale studies (e.g., Auger et al., 2015). For more details see Aumont et al. (2015) where a complete description of PISCES equations along with a brief validation are presented.

PISCES differentiates two phytoplankton functional types: diatoms and nanophytoplankton. The parameterization of diatoms differs from nanophytoplankton in their requirements of silicate (Si), an increased consumption of iron (Fe), and a higher level of nutrient saturation due to its larger size. Both Chl-$a$ content on nanophytoplankton and diatoms are parameterized using the photo-adaptative model of Geider et al. (1997). We refer Chl-$a$ to here as the direct sum of these two compartments, and it will be used as a proxy for primary production. Besides biomass of Chl-$a$, carbon ratios with Fe, and Si (only for diatoms) are explicit prognostic variables of the model. External inputs of iron are specified using forcing data which reproduce iron supplies from rivers, continental-shelf sediments and atmospheric deposition (Tagliabue et al., 2009). Furthermore, PISCES discretises two sizes of zooplankton: micro- and mesozooplankton, and three classes of non-living compartments: the semi-labile dissolved organic carbon pool, and two sizes of particulate organic carbon that differ by their sinking velocities (3 m d$^{-1}$ for small particles, and 50 to 200 m d$^{-1}$ for large particles).

PISCES is coupled on-line to NEMO with a coupling frequency equal to the circulation model time-step (i.e., 40 min). Note that on-line coupling means here one-way forcing of the ecosystem model by the circulation model, since no feed-back of the ecosystem model is taken into account. This strategy of on-line coupling with a maximum frequency is thought to be useful for simulating the ecosystem evolution, while avoiding possible problems brought by the use of averaged physical fields, as in off-line coupling.

A realistic dynamical adjustment of the modeled ocean state is obtained after a 13 years spin-up (1989-2002) starting from the Levitus climatology for temperature and salinity (Levitus et al., 1998). After physical spin-up, the biogeochemical component is initialized in January 2002 from outputs of a global 1/4° PISCES operational simulation performed by MERCATOR Ocean (Elmoussaoui et al., 2011). Between January 2002 and December 2004, a three years spin-up is performed to ensure a consistent biological initial state. After this period, a deterministic simulation of the coupled system, i.e., NATL025-PISCES, is performed for a period of six years, extending from January 2005 to December 2010.

## 2.3 Probabilistic version of the coupled system

Any realistic description of the state of a system involves uncertainties. In the case of coupled ocean models, uncertainties may originate from external forcings (e.g.,the atmospheric data), parameterizations of physical and biological processes that are



not explicitly resolved by the model, omission of unresolved scales, and reduced complexity to limit computational costs. In a previous study (Garnier et al., 2016), two classes of those uncertainties were parameterized to explicitly simulate the errors

associated with the deterministic model formulation; (1) the limitations of the spatial scales resolved by the model, and (2) the simplification of the description of the system to a limited number of state variables and parameters. The first ones were described by following the approach proposed in Brankart et al. (2015), and the second ones were simulated by introducing log-normal stochastic perturbations on seven key biogeochemical parameters.

The stochastic formulation was introduced to produce an ensemble spread that is large enough for building a DA system

while keeping the coupled model stable. A 60 member ensemble simulation for year 2005 was performed by Garnier et al. (2016) who show that the resulting probability distribution (of the annual ensemble simulation) is quite consistent with SeaW-iFS (Sea-viewing Wide Field-of-View Sensor) ocean color observations. Specifically, they assessed the reliability or statistical consistency of the ensemble simulation by comparing it with satellite Chl-*a* data assuming 30% of observation error.

The present study is based on these previous developments, with some additional adjustments to prepare for DA. In partic-

ular, we carried out sensitivity experiments to select an ensemble size that is more costly-efficient, but with the same level of agreement to observations as in Garnier et al. (2016). More explicitly, monthly assimilation experiments with three different ensemble sizes were performed by reducing the original 60 member ensemble to 12 and 24 members. These experiments were assessed by comparing them with the observations used for the assimilation process. We observed the 24 member experiment was capable to conserve the same level of statistical consistency as the original 60 member ensemble while reducing com-

putational costs of the forecast step by up to 60%. Therefore, a total of 24 trajectories of the inherited stochastic simulation developed in Garnier et al. (2016) are used here as the prior PDF for the assimilation problem.

## 2.4 Assimilation scheme

The assimilation system integrates one-day MERIS Chl-*a* observations to daily update the ensemble forecast. The methodology behind this process is based onto a SEEK filter (Pham et al., 1998; Brasseur and Verron, 2006), implemented in the ensemble

system using the System of Sequential Assimilation Modules (SESAM) assimilation platform (Brankart et al., 2012) that deals with all matrix operations required by the assimilation scheme. The assimilation scheme proceeds in two steps. (1) An ensemble forecast in which each ensemble member, i.e., state vector, is propagated forward in time using the full model equations. (2) When a set of observations, i.e., daily swaths of ocean color retrieved from MERIS, is available, the statistical information contained in the ensemble is combined with observations to update the forecasted ensemble. The most relevant aspects of this

second step, referred to as analysis, will be commented below. To propagate the system, the initial condition of the subsequent daily forecast is the updated analysis ensemble obtained by the assimilation of Chl-*a* observations.

The state vector entering the analysis step is composed of all prognostic biogeochemical state variables of the three-dimensional grid following a multivariate approach. To keep the analysis computationally affordable, a prior diagnosis of the multivariate correlations between the observed (Chl-*a* in this case) and non-observed biogeochemical variables have been

carried out. Following the results obtained from this test, 12 out of the 24 biogeochemical state variables are included into the updated state vector. These state variables correspond to nutrients, oxygen, zooplankton, phytoplankton, and Chl-*a* biomass.





The probability distribution of the observed variable, i.e., Chl-*a*, is usually considered as log-normal (Campbell, 1995). A well-known strategy to accommodate the characteristic non-Gaussian distributions of biogeochemical parameters is applying a log-normal transformation (e.g., Ciavatta et al., 2011; Mattern et al., 2017). However, this transformation assumes that the shape of the probability distribution does not change. As this is not often verified, we adopted here another non-linear strategy dependent on the shape of the probability distribution. Anamorphosis transformations (Bertino et al., 2003; Béal et al., 2010) are applied to each variable of the state vector prior to the ensemble analysis step to ensure that marginal PDFs are close to Gaussian. The strategy of these transformations relies on remapping the quantiles of each marginal distribution such that the probability distribution is as close as possible to a Gaussian. These transformations ensure that no value of the variables becomes negative after the analysis update, improve the description of the correlations between Chl-*a* and non-observed variables, and exclude possible causes of breakdown of the simulation. To be compliant with the new variables, observations are also transformed into the anamorphic space defined by the ensemble simulation. After analysis, the corresponding inverse transformations are performed to come back into the original model space and initialize the subsequent daily ensemble forecast.

Relatively small ensembles like the one used here can lead to spurious correlations between distant model grid points. In order to avoid the potential negative effects of these correlations, we employ a domain localization methodology in which a separate analysis for each local domain is applied. In practice, this means that an analysis update is performed for each horizontal grid point, but including all vertical levels and state variables. To ensure continuity between analyses, each analysis uses the observations within a certain localization radius (of one degree in the present case), with an observation error that increases with distance.

## 2.5 Assimilated and independent observations

The observation data set assimilated by our system corresponds to daily swaths of ocean color retrieved from MERIS. Specifically, we use Level-3 binned data accessible at http://earth.esa.int/level3/meris-level3/, that consist on daily-accumulated Level-2 products with standard bin size of 4.6 km. Among other properties, this product provides Chl-*a* estimations (in mg m$^{-3}$) used here to update the ensemble simulation. Additionally, the system performance will be assessed by comparison with ocean color SeaWiFS data accessible through https://oceancolor.gsfc.nasa.gov/data/seawifs/, and with daily surface Chl-*a* fields obtained from the Global Ocean Satellite Observations provided by Copernicus-GlobColour service, and accessible through http://marine.copernicus.eu. This latter product is based on the merging of several sensors (SeaWiFS, MODIS-Aqua, MERIS, VIIRSN, and OLCI-S3A) delivered at 4 km of spatial resolution.

The limited accuracy of ocean color products is taken into account in the assimilation process. Imperfections in the retrieval process of the Chl-*a* concentrations may be due to the presence of chromophoric dissolved organic matter, atmospheric aerosols, or errors in the algorithms at some specific regions, among others (e.g., Gregg and Casey, 2004; Le Fouest et al., 2006). Therefore, a 30% of observational error is considered in agreement with global average standard deviation estimates (e.g., Gregg and Casey, 2004; Mélin et al., 2016).

While inter-comparisons between the data-assimilated simulation and the assimilated observations are necessary to assess the experiment efficiency, the validation strategy is not totally conclusive since they are not completely independent (Gregg





et al., 2009). We thus use an additional independent data set for an objective validation of the assimilation process. Specifically, we use biogeochemical fields extracted from the World Ocean Atlas 2018 (WOA2018; Garcia et al., 2019). The historical *in situ* nutrient measurements available in this data set were produced by the NOOA's (National Oceanic and Atmospheric Administration) National Oceanographic Data Center - Ocean Climate Laboratory as part of the World Ocean Database project

(WOD; Boyer et al., 2013). They will be used to assess the simulation performance on matching the non-observed variables patterns.

## 2.6 Probabilistic validation

Unlike for deterministic simulations, the validation of our DA experiment requires many realizations (members) to be properly evaluated given its probabilistic nature. For that purpose, reliability and resolution scores (see Toth et al., 2003; Candille et al.,

2015, for more information) will be computed from the ensemble. Reliability evaluates the capacity of a model to produce an ensemble probability distribution in agreement with the statistical distribution of a given observation data set. Resolution measures the ability of a model to discriminate distinct observed situations. In other words, the reliability informs of the system's ability to produce PDFs that agree with a given observations' PDF, while resolution informs of the spread of the system's PDFs. These metrics will allow us to measure the skills of our ensemble simulation for predicting the true state of the

ocean biogeochemistry.

To evaluate reliability and resolution, several probabilistic metrics will be employed. We first check the reliability of the DA system by introducing the rank histogram (Anderson, 1996). Rank histograms are computed by sorting all 24 members in ascending order (in the present case according to their Chl-*a* concentration) for each grid point and at a given date. Each observation is then ranked relatively to its location within this sorted ensemble. Observations smaller than the minimum of

the ensemble will take rank '0', while those observations higher than the maximum of the ensemble will take rank 'n'. The statistical consistency of the ensemble can then be evaluated by studying its shape (Candille et al., 2015; Germineaud et al., 2019). Rank histograms may be: (1) flat, which indicates the distribution of the model is accurate with the observations, i.e., perfect reliability, (2) under-dispersed or U-shaped, which indicates the spread of the ensemble is too small (too many observations lay outside the extremes of the ensemble), or (3) over-dispersed or dome-shaped, which indicates the spread of

the ensemble is too large (too many observations lay near the center of the ensemble).

For measuring the resolution of the system, and obtain a full probabilistic validation of the ensemble, we use the continuous rank probability score (CRPS). Let *x* to be a parameter of interest (Chl-*a* in our case) to which corresponds a real observation. Then, CRPS corresponds to the distance between the simulation and the observation, as defined in

$$CRPS = E\left[\int_{\mathbb{R}} (F_p(x) - F_o(x))^2 dx\right]$$

where $E$ is the mean over all observations at a given date, and $F_p(x)$ and $F_o(x)$, the cumulative distributions of the model and the observations.





CRPS can be decomposed as the sum of the ensemble reliability (Reli) and the ensemble potential resolution (Resol), i.e., the resolution in the case of perfect reliability (see Hersbach, 2000, for more details).

$$CRPS = Reli + Resol$$

According to CRPS, a skillful probabilistic system must satisfy two criteria: Reli should be null, and Resol must tend to zero and, in any case, much inferior than the reference value of the CRPS when it is only based on the reference data set (without data assimilation).

## 3   Results

It is instructive to evaluate how the assimilation process affects the original ensemble simulation. In this section, the impacts
of the assimilation on the variability in space and time of key biogeochemical parameters are assessed by comparing the assimilated experiment with an analogous 24 member ensemble of model simulations in which no observational data have been assimilated. This ensemble run is performed over the same period, and referred to as 'free run'.

### 3.1   Skill on reproducing surface Chl-*a*

Both the assimilated and the free run simulations are compared to daily surface Chl-*a* fields obtained from the Global Ocean
Satellite Observations (see section 2.5). Merged satellite products are selected here to diminish missing data due to cloud cover and still resolve mesoscale spatio-temporal variability. For the assimilated simulation, the analysis step, which corresponds to the ensemble computed after the update, is shown. Surface Chl-*a* daily composites are presented for three different dates; 19th April 2005, 15th May 2005, and 5th October 2005 (*Fig.* 2). These dates are selected for representing contrasting periods. The first two dates coincide roughly with the well documented spring-bloom period of the North Atlantic. The availability of light
and nutrients during this period drives phytoplankton growth, which leads to relatively high values of Chl-*a* at surface. The last date represents a period after summer when conditions change due to the reduction of sunlight over the surface.

    The large-scale spatial distribution of surface Chl-*a* is captured by the DA simulation. High Chl-*a* values at regions such as the Gulf Stream, the North Sea, the Amazonian delta, and the western coast of Africa, are successfully reproduced during the first days of the experiment (*Fig.* 2 top panels). However, a too strong gradient between the oligotrophic conditions of the
North Atlantic subtropical gyre and temperate waters northwards is shown. The ensemble median of the free run experiment displays a stronger gradient, indicating that the assimilation of surface Chl-*a* data slightly improves this situation.

    About a month later (*Fig.* 2 middle panels), the inferred large-scale picture of surface Chl-*a* distribution remains close to that displayed by satellite observations. The bloom of Chl-*a* in temperate waters is well reproduced by the DA simulation both attending to magnitude and geographical location. Upwelling areas along with other zones with high Chl-*a* concentrations
are also well depicted showing a good performance to match highly productive regions. By contrast, the gradient northwards of the oligotrophic open ocean is too pronounced, showing no transition between both regimes as evidenced in the satellite


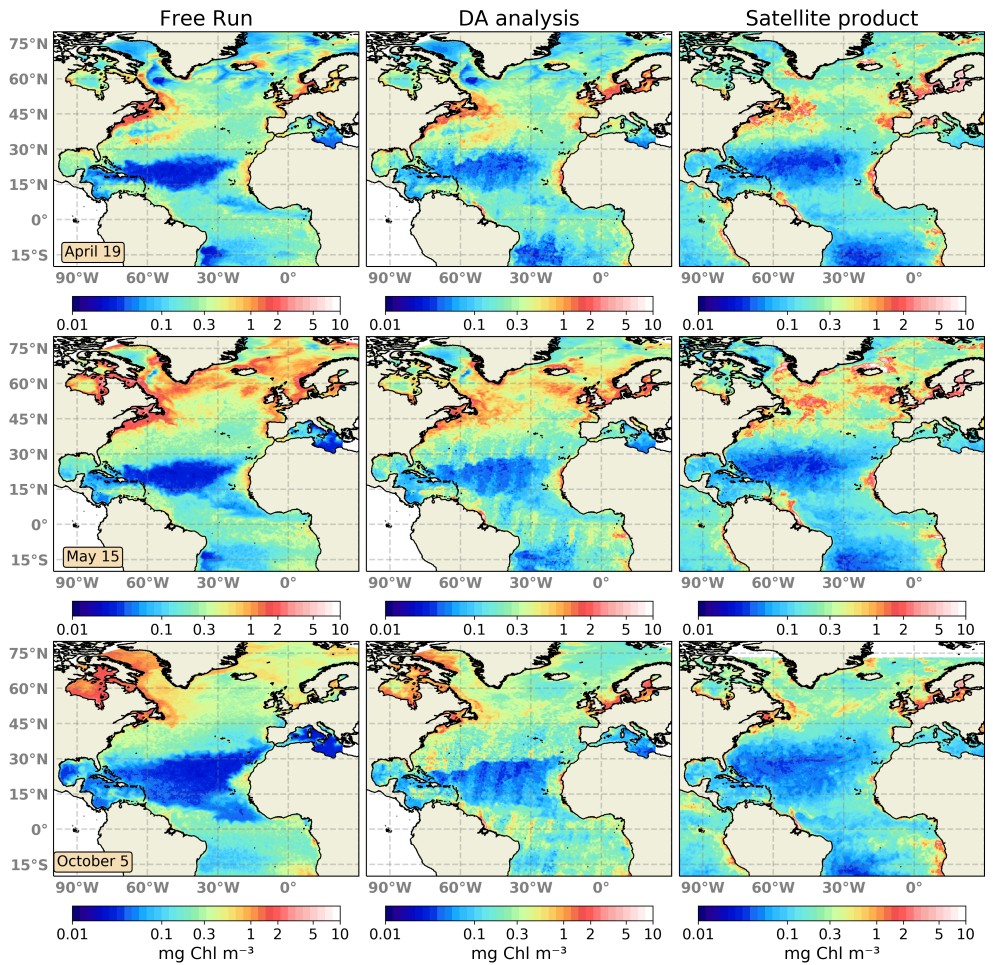

**Figure 2.** Surface maps of Chl-*a* (mg Chl m$^{-3}$) for 19th April 2005 (top panels), 15th May 2005 (middle panels), and 5th October 2005 (bottom panels). The ensemble median of the non-assimilated free run experiment (left panels), the analysis ensemble median of the assimilation experiment (middle panels), and merging daily surface Chl-*a* fields obtained from the Global Ocean Satellite Observations (right panels) are represented.

map. Moreover, corrections made by daily satellite swaths leave imprints of their trajectory in the analysis map. The free run overestimates Chl-*a* in high-latitude regions while it underestimates it within the North Atlantic subtropical gyre.

After the summer period, at the beginning of October 2005 (*Fig. 2* lower panels), the Chl-*a* distribution changes; regions with

the highest concentrations relaxed their values, while concentrations at the open ocean slightly increased. The representation of surface Chl-*a* degrades during this period. Concentrations within the subtropical gyre agree with observations, but gradients both at its northern and southern boundaries are too strong. In this transition zone, inferred values double the concentrations displayed by observations. In the rest of the domain, the Chl-*a* pattern improves that showed by the non-assimilated simulation.





The free run experiment exhibits a strong overestimation at the Gulf Stream region as already observed by Garnier et al. (2016)
with a 60 member non-assimilated version of the coupled model. Moreover, oligotrophic conditions at the subtropical gyre are
too low, which may indicate the spread of the ensemble is unable to capture the whole observation's variability.

## 3.2    Probabilistic regional assessment

An ensemble system should be statistically consistent with observations in order to be objectively considered as realistic. To
evaluate the reliability metric, we present rank histograms (*Fig.* 3) computed for each grid point by accumulation over all 24
members of the ensemble at three different Longhurst provinces (Longhurst et al., 1995, *Fig.* 1 provides the location of the
provinces). Both the assimilation analysis and free run ensembles are displayed. Ranks are computed against SeaWiFS ocean
color data extracted for the same day with a considered 30% of observation error. In practice, the latter error means that for
each realization a Gaussian white noise with a standard deviation of 30% of the satellite Chl-*a* concentration is added to each
ensemble member.

Histograms are good indicators of how the assimilation of surface Chl-*a* affects the probability distribution of the ensemble.
The histogram for province 4 (*Fig.* 3 first row), which corresponds to a major part of the eastern North Atlantic temperate waters
($\sim$40-60°N; $\sim$10-45°W), displays the good performance of the non-assimilated simulation reproducing the given observations.
The histogram is flat except for a slightly tall rank '1' that indicates the highest observations are not included in the spread of
the ensemble. When observations are assimilated, the system preserves its reliability but modifies the shape of the histogram.
The distribution of ranks flattens showing a better reliability. The shape of the histogram illustrates the ensemble is now able
to include the highest values of Chl-*a*, though few ranks accumulate in the left side of the histogram.

An accumulation of ranks forms a dome in the middle of the histogram for province 7 (second row), which corresponds to
the southern boundary of the North Atlantic subtropical gyre ($\sim$13-26°N; $\sim$16-75°W). The free run surface map (see *Fig.* 2
middle panels) showed too low values in the northern part of the province, and too high values in the southern part, that differ
from the smooth gradient showed by the observations. When ocean colour data is assimilated, ranks' distribution becomes
more homogeneous. A moderate dome of ranks in the right side of the histogram still appears, yet the envelope of the ensemble
reconciles well with the given observations.

The histogram of the free run shows an accumulation of ranks at the left side, i.e., too many ranks '0', for province 18
(third row). This is the eastern branch of the subtropical gyre of the North Atlantic that goes roughly from the center of the
Atlantic to the east European and African coasts (from 30 to $\sim$44°N). The surface map (*Fig.* 2 middle panels) showed an
overestimation of Chl-*a* for most of the region, except for the oligotrophic center of the subtropical gyre where values were
almost negligible. When observations are assimilated, the values of the oligotrophic area increased while values closer to the
coasts tended to diminish. These corrections are also reflected into the histograms by a re-distribution of the lowest ranks to
the right. Nonetheless, improvements are limited, and there is an overpopulated left side of the histogram.

Lastly, the right branch of the North Atlantic subtropical gyre is included within province 6 ($\sim$30-40°N; $\sim$30-75°W; last
row). In this area, the free run histogram shows a strong accumulation of ranks at the left extreme, i.e, the ensemble systemati-
cally overestimates observations (positive bias). This under-dispersed shape indicates that the ensemble is unable to cover the

**Figure 3.** Surface Chl-*a* rank histograms of the 24 member free run experiment (magenta; left panels) and the 24 member analysis ensemble assimilation experiment (light green: right panels), in comparison with SeaWiFS data for 15th May 2005. A 30% SeaWiFS observation error is taken into account. Longhurst provinces 4, 7, 18, and 6 are represented.

lowest observations. In this case, the assimilation of satellite data is unable to improve the reliability of the system. Moreover, the accumulation of lower values not included within the probability distribution of the ensemble increases after assimilation.



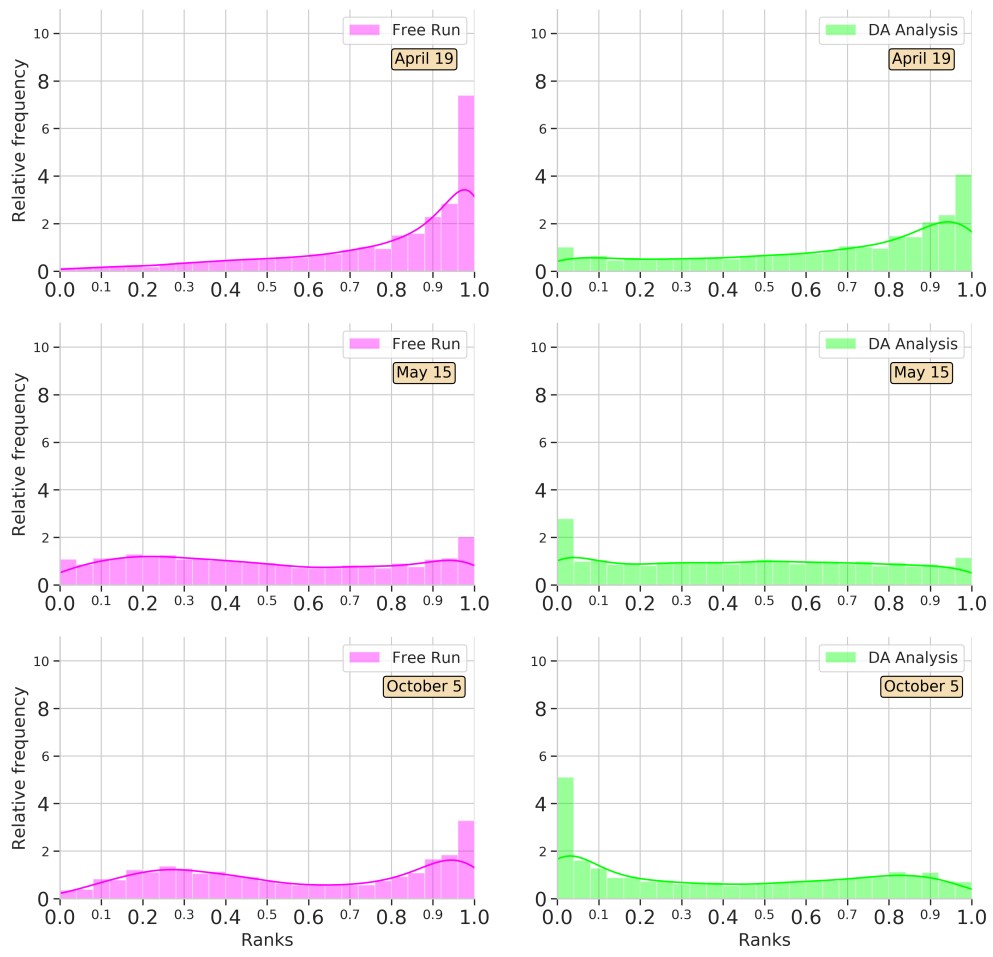

**Figure 4.** Surface Chl-*a* rank histograms of the 24 member free run experiment (magenta; left panels) and the 24 member analysis ensemble assimilation experiment (light green: right panels) in comparison with SeaWiFS data for province 4. Ranks are computed for 19 April, 15 May, and 5 October 2005. A 30% SeaWiFS observation error is taken into account.

To see the time evolution of the reliability of the ensemble in province 4, *Fig.* 4 shows rank histograms computed at three different periods. A week after the initialization of the experiments, i.e., 19 April 2005, there is an accumulation of ranks in the right side of the free run histogram (negative bias). The assimilation of satellite information redistributes ranks to the left. Yet there is still an underestimation, the probability distribution of the analysis ensemble fits better with observations thus decreasing the bias. A month after, as we observed in *Fig.* 3, both the free run and DA simulations display flat histograms

indicating a good performance of the system. In particular, the system reproduces the increasing on Chl-*a* that occurs during the spring bloom period, which takes place around this date in the province (e.g., Follows and Dutkiewicz, 2001). In October,



the free run tends to accumulate ranks on the right side, while an accumulation on the left side of the histogram is depicted for DA analysis. Notwithstanding, the distribution of ranks is more homogeneous after the assimilation process.

To complement reliability measurements, we present an analysis of the CRPS metrics for an in-depth evaluation of the assimilation effects. Using all daily satellite observations available during the simulation period, we calculated the Reli and Reso terms of the CRPS decomposition for provinces 4 and 6; two provinces with contrasted behaviours. Rank histograms (*Fig.* 3) showed the ensemble is consistent with observations in province 4 while it underestimates them in province 6. Similarly, the reliability term of the CRPS metric (*Fig.* 5a) shows the improvements (closer to zero) made by the assimilation process on province 4, in which the prior probability distribution was already coherent with observations. This pattern, however, reverses

around August when the integration of data deteriorates the metric. This situation lasts until mid-December, when reliability for both simulations begin to coincide each other until the end of the experiment. By contrast, reliability of the free run simulation is generally closer to zero for province 6 during the whole experiment.

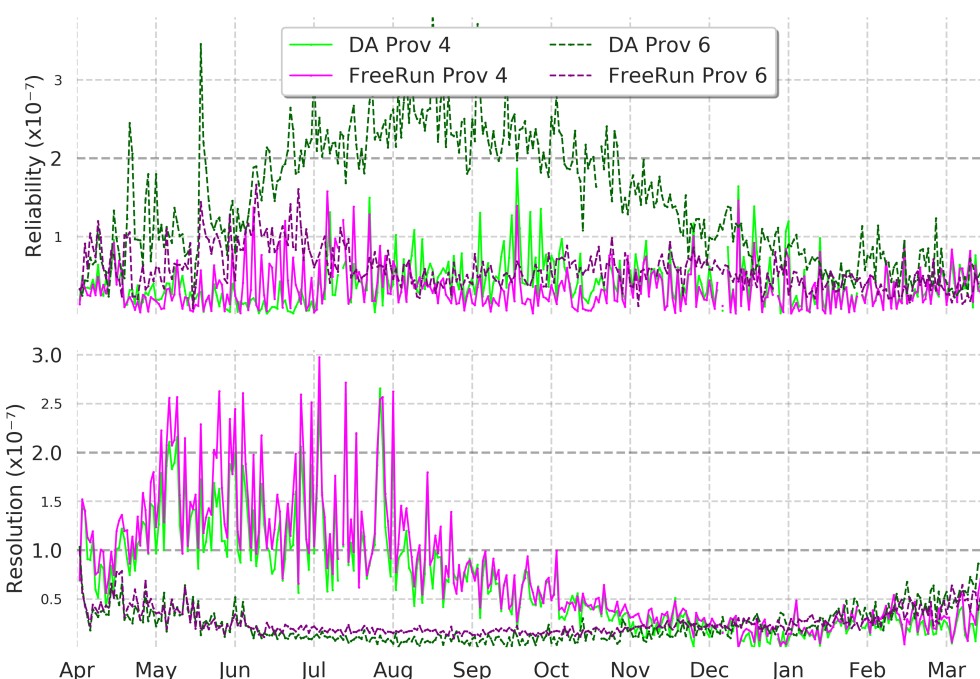

**Figure 5.** Time series (6th April 2005 to 5th April 2006) of reliability (upper panel) and resolution (lower panel) computed from CRPS decomposition for the 24 member free run (in magenta) and the 24 member forecast ensemble assimilation (in light green) experiments. Longhurst provinces 4 and 6 are represented.

  Time series of the resolution part of CRPS (*Fig.* 5b) show the metric tends to zero for both systems, indicating a good global performance. As expected after precedent metrics diagnostics, the analysis update generally improves the resolution for

province 4. During summer, the resolution of both simulations increases until the end of the season.



### 3.3 Assessment of the multivariate scheme

The multivariate scheme employed here allows corrections on surface Chl-*a* to extend to other variables. Considering a complex model such as PISCES, these changes may provoke several variables to no longer satisfy model equations, and thus results produced by these adjustments should be assessed. Moreover, surface corrections both in the observed and non-observed

quantities are projected vertically in the water column thus altering the vertical structure of the water column. In order to evaluate the balances between Chl-*a* and those variables that have important relations with it such as nutrients, monthly means of nitrate and phosphate extracted from the WOA2018 data set are compared with data depicted by our simulations. Specifically, vertical profiles of Chl-*a* and nutrient concentrations at two points placed at regions 4 and 6 are presented (*Fig.* 6).

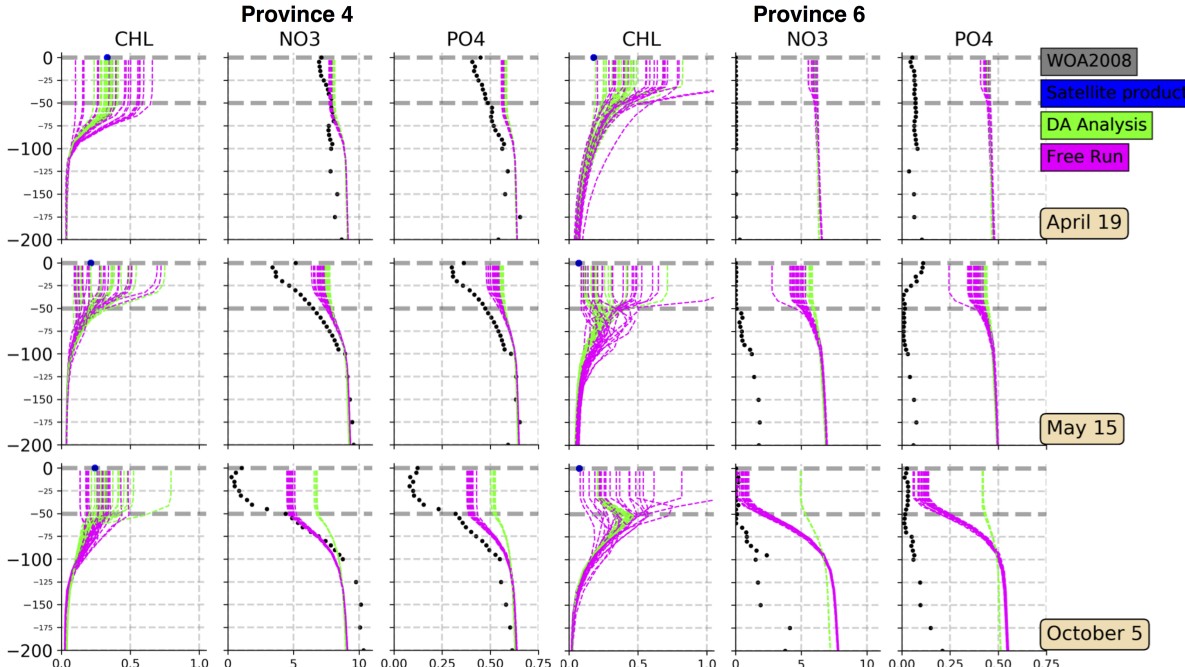

**Figure 6.** Vertical profiles (0-200 m) of Chl-*a* (mg Chl m$^{-3}$), nitrate (mmol N m$^{-3}$) and phosphate (mmol P m$^{-3}$) for province 4 (left panels; at 35° N, 60° W) and province 6 (right panels; at 50° N, 15° W) for 19th April 2005, 15th May 2005, and 5th October 2005. The 24 member free run (in magenta) and analysis (in green) ensembles are represented. Black dots correspond to monthly mean nitrate and phosphate concentrations extracted from WOA2018 database. Blue dots correspond to daily mean surface Chl-*a* obtained from the Global Ocean Satellite Observations.

      Vertical profiles show that both ensembles are capable of displaying a wide range of Chl-*a* values within the first meters

of the water column. As expected, the spread of the analysis reduces while the subsequent forecast (not shown) will restore it accordingly to match the satellite's uncertainty for the next update. The envelopes of both simulations decrease towards the bottom of the mixed layer. From there, concentrations displayed by both simulations coincide. This indicates the extension up



to which surface corrections are projected into the vertical. The spread of the ensemble reduces when nutrients are represented. In general, the assimilation process increases their concentrations within the mixed layer.

In province 4 (left panels on *Fig.* 6), the concentrations of nutrients in the mixed layer decrease over time. In October, mixing is close to its lowest (Zhang et al., 2018) and so are nutrients' concentrations. Inferred nutrients follow the seasonal pattern of decreasing towards October. However, their values are relatively high in comparison with climatological data. The assimilation process tends to further increase nutrient's availability within the mixed layer, yet being capable of correctly simulating surface Chl-*a*.

The water column is poor in nutrients at province 6 (right panels in *Fig.* 6). However, both simulations show their concentrations to be up to seven times higher than WOA data. As observed for province 4, their concentrations decrease towards the end of the summer. The free run simulates this decreasing, especially during October when concentrations are close to observations. By contrast, the analysis moves away the distribution of nutrients from climatology. Corrections made by surface information are unable here to include the given Chl-*a* observations. Surface data is overestimated by both ensembles. The

assimilation process approaches the ensemble to observations in the first cycles of assimilation, but it strongly overestimates them in October. The free run shows a too wide spread that reproduces Chl-*a* concentrations up to an order of magnitude higher than satellite data.

### 3.4 Impact on the subtropical region

Figures presented in precedent sections indicate an erratic behaviour of the system representing the transition zone between the

oligotrophic subtropical area and temperate waters northwards. In order to illustrate the vertical distribution of biogeochemical properties before and after assimilation in this area, we consider meridional vertical sections of Chl-*a* crossing the subtropical gyre and temperate waters at 45° W with nutrients (nitrate + ammonium) isolines superimposed (*Fig.* 7). The same three dates used before are represented.

During April (*Fig.* 7a), high values of Chl-*a* deepen up to ∼50 m depth in both experiments. However, the oligotrophic region

reaches further north after the assimilation due to a deeper nutrient-depleted subsurface layer south of ∼30° N. The deep Chl-*a* maximum (DCM) of the subtropical region is placed below 100 m depth in both simulations in agreement with observational studies (Pérez et al., 2006). After assimilation, the DCM is disconnected from the subsurface maximum of temperate waters by the vertical slumping of nutrients isolines.

A horizontal strong gradient of nutrient isolines is observed in the DA analysis section of May (*Fig.* 7b). Several patches of

high Chl-*a* values are evident south of ∼35° N, from where the water column setting becomes similar to that displayed by the free run simulation. These patches may be caused by the vertical propagation of surface corrections. By contrast, the free run simulation shows a more logical distribution of parameters in which high Chl-*a* waters are related to nutrients' availability.

During October (*Fig.* 7c), differences after assimilation are more noticeable. In this period, the vertical distribution of nutrients has a key role controlling phytoplankton growth in the region (e.g., Dutkiewicz et al., 2001), and concentrations of

Chl-*a* are relatively low as the nutricline is deep enough to limit production. Since the free run overestimates Chl-*a* during this period in the region, the assimilation process reduces its concentrations. As a consequence, nutrients accumulate in the first

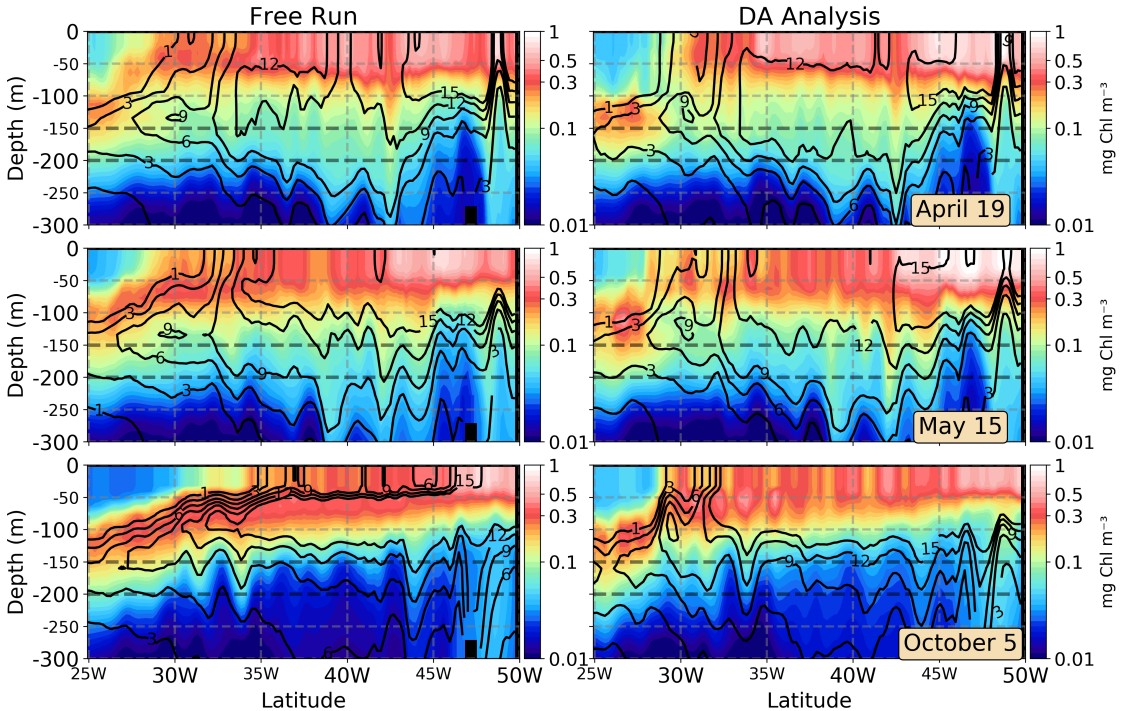

**Figure 7.** Meridional vertical sections (0-300 m) of Chl-*a* (mg Chl m$^{-3}$) at 45° W, 25 to 50° N, for 19th April 2005, 15th May 2005, and 5th October 2005. The ensemble median of the free run (left panels) and the assimilated (DA Analysis; right panels) simulations are represented. Nitrate+ammonium isolines (mmol N m$^{-3}$) are included in solid black lines.

100 m of the water column north of ~30° N after the update, and destabilize the equilibrium between the biomass of producers (that decreases) and the availability of nutrients (that increases). Since biogeochemical dynamics is highly dependent on this equilibrium, subsequent forecasts lead to a rapid increase of Chl-*a*, and into a severe overestimation over time that reduces the
extension of the oligrotrophic region to the north.

## 4 Discussion

### 4.1 Impact of the assimilation in the observed variable

The North Atlantic ocean is a complex basin that includes a large number of biogeochemical regimes (see *Fig.* 1) that make it difficult to simulate using an holistic modelling system (DeYoung et al., 2004). By employing DA, we aim to reduce the impacts
of model errors on the representation of ocean biogeochemistry by combining model information with available observations (Gregg et al., 2009; Ciavatta et al., 2011; Ford and Barciela, 2017). Accordingly, surface maps presented in *Fig.* 2 show that the assimilation process reduces the discrepancies between satellite observations and the non-assimilated free run experiment





over a major part of the domain. In particular, the DA simulation better represents values and geographical location of some structures and events such as the spring bloom period, the Gulf Stream, or phytoplankton fronts. By contrast, the assimilation

scheme appears to be unable to deal with an unrealistic too abrupt front that separates the oligotrophic and temperate waters conditions (from about 25 to 35°N).

By using rank histograms, we evaluate the capability of the assimilated and the free run ensembles to agree with observations on selected Longhurst provinces (see *Fig.* 3). Histograms illustrate that the response of the system to the assimilation of satellite data depends upon the reliability of the prior ensemble. The assimilation process improves the statistical consistency of the

system where the free run probability distribution is homogeneous as in province 4. The DA process also enhances reliability in those regions where the shape of the free run histogram is over-dispersed as in province 7. In these regions, the stochastic parameterization is enough to describe properly the variability of the system, and only a relatively small percentage of the observations lie outside of the limits of the ensemble (∼10%). Then, the assimilation process makes use of this information to increase the model skills both by reducing the dispersion and by redistributing ranks to a more homogeneous shape. The

redistribution of the ensemble also raises its resolution showing that the posterior ensemble better describes a wide variety of biogeochemical situations (see lower panel of *Fig.* 5).

By contrast, in regions where the prior probability distribution is strongly under-dispersed as in provinces 6 and 18, the assimilation of satellite information is unable to raise the reliability. Since corrections are computed in the range explored by the prior ensemble, the assimilation scheme cannot correct prior distributions that exclude the full variability of the observations.

In these provinces, the spread of the prior ensemble is insufficient to represent the range displayed by the observations; the ensemble consistently overestimates them during the annual cycle, and so ranks accumulate at the left extreme of the histograms (positive bias). These two provinces occupy a major part of the oligotrophic subtropical gyre of the North Atlantic where Chl-*a* is generally low during the whole year. Since Chl-*a* values can never become negative, the random perturbations introduced into the model formulation to create a probabilistic simulation (Garnier et al., 2016) preferentially induce to increase Chl-*a*

concentrations. An inferior boundary that may cause this overestimation.

The inability of the assimilation process to improve the skills of the simulation in these latter regions points out the necessity to appropriate define the stochastic parameterizations of the prior PDF as a prerequisite to use DA. Particularly, uncertainties should be described accordingly with the biogeochemical characteristics of each region in order to include a major part of the observation variability.

## 4.2    Non-observed variables

The effects of the assimilation process to unobserved variables is a major issue in biogeochemical DA (e.g., Rousseaux and Gregg, 2012; Ciavatta et al., 2018). Several studies (e.g., Ciavatta et al., 2011) have found that the integration of surface ocean color may cause problems in the nutrients vertical distribution when model's equation are not 'plastic', in the sense of constraining the ability of the assimilation process to correct the inferred variables. In this regard, it is important to notice that our

multivariate analysis scheme allows corrections on five nutrients, the Chl-*a* content of each phytoplankton group, phytoplankton and zooplankton biomasses, and oxygen, while it only uses Chl-*a* satellite data to constrain the ocean biogeochemistry. It





is plausible that modifications on biogeochemical variables would make some of them not to comply with the governing model equations anymore. Particularly in those regions where the model is not plastic enough to absorb modifications on the tight correlations between observed and unobserved state vectors. In some cases, these modifications may develop into simulation instabilities that can lead subsequent forecasts to deteriorate both the observed and unobserved variables (Ciavatta et al., 2011, 2018; Gregg et al., 2009). For instance, large discrepancies between observations (high concentrations of Chl-*a*) and the model (lower concentrations) in Gregg (2008) caused their model to become unstable due to nutrient depletion. In the present case, the assimilation process has two effects on the vertical distribution of nutrients (see *Fig.* 6 and *Fig.* 7): (1) it reduces significantly the spread of the ensemble, and (2) it tends to increase their concentrations within the first ∼100 m depth.

Nutrients were not perturbed by the stochastic parameterizations (see Garnier et al., 2016) and so increases during the analysis update cannot be attributed but to the assimilation process. In the northern region of the North Atlantic subtropical gyre, Chl-*a* is overestimated by the prior ensemble and so surface corrections preferentially reduce their concentrations. Since nutrients are negatively correlated with the observed variable, the corrections made by the assimilation process would increase nutrient's availability. Ourmières et al. (2009) observed that the distribution of nitrate controls the biogeochemical dynamics of the subtropical region by employing physical - only , biogeochemical - only (nitrate data), and physical - biogeochemical combined assimilation techniques over a coupled system. As a consequence, the amount of nutrients available in the water column after the analysis would alter the correlations with the observed variable during the subsequent forecast in this area. If the ensemble spread were correctly established in the region, PISCES equations would be capable to absorb these corrections. However, the ensemble is not stochastic enough in provinces 6 and 18, and the increasing of nutrients lead to a consistent overestimation of Chl-*a*. By contrast, in the rest of the domain, the parameterizations of the uncertainties are consistent with observations, and extrapolation of the assimilated information to non-observed variables works correctly. The assimilation process increases the reliability of the ensemble, and the information spreads appropriately to the rest of the variables. As a result, the subsequent daily forecast is based onto a more homogeneous ensemble improving its general performance.

### 4.3 Assimilation on fluctuations

The precedent section has shown that, despite the stochastic parameterization, PISCES equations are still not plastic enough in the region north of the subtropical gyre as to absorb corrections made by observations. A possible way to alleviate these inconsistencies would be to remove part of the modifications made by the assimilation process. With that aim, we implement a methodology that aims to apply the assimilation process only to the fluctuation part of the coupled model. It consists in performing time-independent transformations to both the ensemble forecast and the observations prior to the analysis update (see schematics on *Fig.* 8). For that end, we compute their climatologies, and use them to separate between the climatological and the fluctuating components of the system, applying the assimilation only to the latter. For the sake of clarification, climatology corresponds to the marginal probability distribution of a considered variable for each specific location compiling all times. In other words, fluctuations are related to the rank of a given value within the climatological distribution.





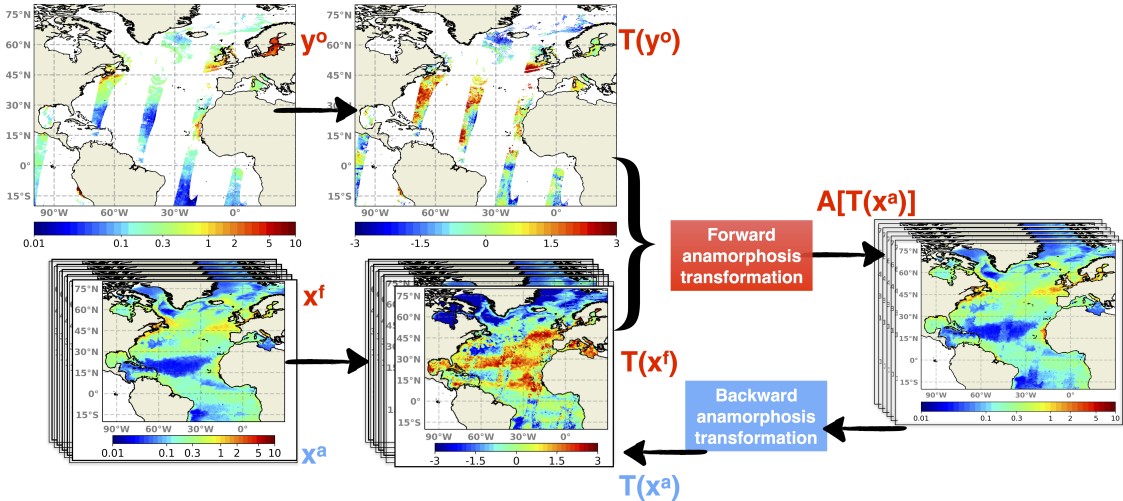

**Figure 8.** Schematics of the assimilation process including time-independent transformations. Both daily ensemble forecast ($X^f$) and daily satellite observations ($y^o$) are transformed by their own climatologies. Then, they are projected into the anamorphic space to deal with non-gaussian distributions prior of entering the analysis step. After the update, the ensemble analysis ($X^a$) is converted back to initialize the subsequent daily forecast.

These transformations would smooth out the influence of the assimilation process on those regions where corrections provoke
strong changes. In addition, this method would presumably increase the agreement between observations and the probability
distribution of the forecast ensemble by normalizing their marginal PDFs by using their climatologies.

A one-month assimilation experiment (TrDA) using this methodology is performed. In order to illustrate the effect of trans-
formations on the ensemble simulation, vertical profiles of Chl-*a*, nitrate, and phosphate are presented for 5th May 2005 (*Fig.*
9), the last day of the experiment. Profiles are placed at province 6 since the methodology aims to reduce the inconsistencies
between observed and non-observed variables found in this region. Profiles illustrate that the transformed ensemble keeps the
values displayed by the non-transformed simulation, while it increases the envelope of the ensemble by reproducing lower
values.

When climatologies are taken into account, the increasing on the concentrations of both nitrate and phosphate after cor-
rections is reduced. Though these changes are insufficient to include observations at this specific position, they indicate that
reducing the effect of the assimilation in regions where the plasticity of the model is insufficient diminishes the inconsistencies.

## 5    Summary and conclusions

Satellite-derived surface Chl-*a* data are daily assimilated into a three dimensional 24 member ensemble configuration of a
coupled NEMO-PISCES model for the North Atlantic. As shown throughout the text, the assimilated system has brought
us promising results. A regional diagnosis of a one-year assimilation experiment has revealed that the integration of surface



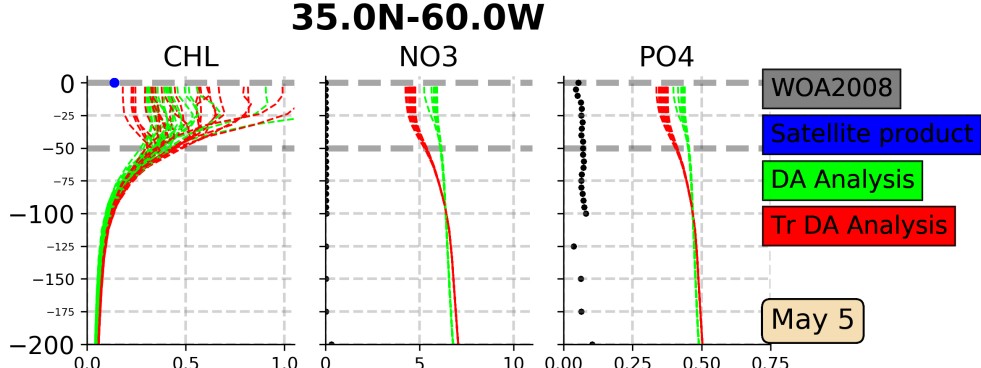

**Figure 9.** Vertical profiles (0-200 m) of Chl-*a* (mg Chl m$^{-3}$), nitrate (mmol N m$^{-3}$) and phosphate (mmol P m$^{-3}$) at province 6 (50° N, 15° W) for 5th May 2005. The 24 member analysis (in green) and the 24 member transformed analysis (in red) ensembles are represented. Black dots correspond to monthly mean nitrate and phosphate concentrations extracted from WOA2018 database. Blue dots correspond to daily mean surface Chl-*a* obtained from the Global Ocean Satellite Observations.

information increases the skills of the ensemble system in a major part of the model grid when compared to an analogous non-assimilated free run simulation. Particularly, the assimilation of satellite data improves the representation of the surface Chl-*a* variability both in location (upwelling areas, subtropical gyre, Gulf Stream, etc) and seasonality (spring bloom, winter mixing, etc). Therefore, the stochastic parameterizations introduced into the system by Garnier et al. (2016) have shown to be adequate for undertaking DA in most of the considered domain. Where the prior ensemble includes the variability shown by

the observations and their uncertainties, the assimilation process improves its probability distribution increasing the agreement with observations (reliability), and its capability to display different community behaviours (resolution). Moreover, corrections are appropriately transferred to unobserved state vectors by the multivariate scheme.

In the northern region of the North Atlantic subtropical gyre, however, the multivariate corrections produce values that are often inconsistent with model dynamics, which can affect the correlations between the biogeochemical variables. In this region,

the simulation cannot absorb adequately the corrections brought by the observations, i.e., the simulation is not plastic, and the system's performance deteriorates after the assimilation process. Particularly, the analysis update increases the concentrations of nutrients producing instabilities that lead the subsequent forecast to degrade biogeochemical fields. These results suggest that the description of uncertainties needs to be refined according to the biogeochemical characteristics of each Longhurst province.

One possible approach to reduce these instabilities would be to relax the assimilation effects on those areas. Therefore, we carried out a experiment in which corrections are only applied to the fluctuation part of the model. For that end, we apply transformations both to observations and the forecast ensemble before entering the analysis update using their climatologies. Results from a one-month experiment show that these transformations reduce the strong effects of the assimilation increasing nutrients concentrations in the region that lead to inconsistencies.





Including information of biogeochemical fields in the water column into the assimilation scheme would also improve the representation of the biogeochemical state of the ocean. *In situ* information would be thus explicitly included by the system at depth. Nowadays, the only sources of such measurements are limited to the prospects of BIO-Argo floats (Claustre, 2009; Xing et al., 2012). Terzić et al. (2018) assimilated BIO-Argo information into a one-dimensional model and improved the DCM spatial and seasonal representation. Cossarini et al. (2019) succeed in improving the Chl-*a* depiction over the Mediterranean

Sea by assimilating vertical Chl-*a* information supplied by BIO-Argo floats. These recent works open a horizon to constrain biogeochemical model simulations from vertical information. However, at basin scales, the current state of the network allows it to be used for validation purposes, but their limited spatial coverage makes them unusable for assimilation procedures. In the other hand, a possibility to include nutrients information is to introduce synthetic information (e.g., Xiao and Friedrichs, 2014; Yu et al., 2018) from a non-perturbed analogue simulation. In this direction, synthetic observations would be important

in future efforts heading the improvement of biogeochemical data assimilation systems.

*Acknowledgements.*  This work was conducted as a contribution to the GLO-HR-ASSIM project, funded by the Copernicus Marine Environment Monitoring Service (CMEMS). CMEMS is implemented by Mercator Océan International in the framework of a delegation agreement with the European Union. Additional support for this study was also provided by the CNES/OSTST/MOMOMS project. The calculations were performed using HPC resources from GENCI-IDRIS (grant 2018-011279).





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
