# Peer review of "Assimilation of chlorophyll data into a stochastic ensemble simulation for the North Atlantic Ocean"

_Ocean Science, 2020_

## Referee Comment (RC1) · Anonymous Referee #1 · 25 Mar 2020

This study presents an ensemble data assimilation scheme for ocean colour, based on stochastic parameterisations and the SEEK filter. A 24-member ensemble is run for one year with and without assimilation, and assessment made of the ensemble spread and fit to observations. The assimilation generally improves both of these, as long as the prior ensemble spread is sufficient. If not, then the assimilation can degrade unobserved variables. This impact was reduced in a short experiment where the assimilation was applied only to anomalies from model climatology.

The paper is interesting and well written, and the assessment clear and balanced. A few things need expanding on or clarifying, as detailed below, but if those are addressed then I recommend publication in Ocean Science.

Major comments:

Given that the paper is taking a previously used deterministic assimilation scheme (e.g. Fontana et al., 2013) and turning it into an ensemble scheme, I was surprised that no comparison was made to a static implementation of the SEEK filter. I appreciate that the main focus of the paper is to study the ensemble aspects, and that ensembles give probabilistic information that is more widely useful, but given the 24-fold increase in computational cost, it would be useful to see how the ensemble median compares to a deterministic assimilation run. A deterministic run is mentioned in the text (lines 98-99) but not presented, so hopefully this would not involve too much extra effort. It could either be an extra sub-section of the results, or incorporated into some of the existing figures.

The last paragraph of Section 2.3 briefly states that ensemble sizes of 12, 24 and 60 members were compared, and 24 able to give similar results to 60. The issue of ensemble size is an important one that will be of wider interest, so I think this assessment should be presented in the paper.

Section 4.3 is very interesting but also brief. It's fine for it to just be a one-month experiment, but it would be useful to expand on both the methodology (e.g. is the seasonal cycle considered in calculating the climatology?) and the assessment (what's the general impact on chlorophyll skill?).

Minor comments:

Figures: Many use a rainbow colour scheme, which is increasingly discouraged (e.g. Hawkins et al., 2015; https://doi.org/10.1038/519291d). There is no "best" colour scheme I can recommend, but it is worth considering if there is a more appropriate colour scheme for these plots.

Figure 1: It would be best to mask out areas which are not in the model domain (e.g.
the Pacific and eastern Mediterranean).

Line 65: "eddy-resolving" should be "eddy-permitting".

Line 67: "ERA-INTERIM atmospheric fields (Brodeau et al., 2010)." The reference refers to ERA40, not ERA-Interim.

Lines 98-99: "a deterministic simulation  $\dots$  for a period of six years" – this doesn't seem to be presented?

Lines 107-108: Please provide a little more detail on the perturbations, so the casual reader doesn't need to read the references.

Line 163: Worth clarifying that only SeaWiFS, MODIS and MERIS are used for 2005.

Line 170: Remove "completely".

Line 173: "NOOA" should be "NOAA".

Line 232: "satellite swaths leave imprints of their trajectory". It would be good to discuss why this might be happening in the Discussions section. Is it due to the 1 degree localisation radius? Does it imply that the increments are not being retained by the model?

Lines 254-255: "preserves its reliability ... showing a better reliability." I understand what's meant, but it's maybe worth rephrasing these two sentences to be clear about how the reliability has/has not changed.

Line 293: "the metric tends to zero". Perhaps I misunderstand what's meant, but it looks to me like this is a seasonal feature, and the resolution is starting to increase again the following spring, rather than it tending to zero and staying near zero.

Figures 6 and 9: In the labels, the black text on dark blue for "Satellite product" is very hard to read, and "WOA2008" is grey in the label but black on the plot. I think it worth altering how the labels are plotted for clarity. Also, what's the reason for the dotted grey
line at 50m depth?

Figure 7: The x-axes should be labelled with "N" rather "W". What is the reason for the bold dotted lines at 150m and 200m?

Line 375: "An inferior boundary that may cause this overestimation." I think this needs expanding on.

Line 404: "the ensemble is not stochastic enough". I think a more accurate phrasing would be "the ensemble has insufficient spread" or something similar.

Line 409: "a more homogeneous ensemble". Again, I think this needs rephrasing. The histogram is more homogeneous, which means the ensemble has more appropriate spread, rather than being homogeneous itself.

Lines 450-454: Is the assimilation just having a weaker impact, or is it having a better impact due to the reduction of model bias in the assimilation?

Line 462: "unusable". Arguably, but I would suggest "of limited use" or "insufficient".

OSD

---

## Referee Comment (RC2) · Anonymous Referee #2 · 23 Apr 2020

This paper describes the assimilation of chlorophyll into a model of the North Atlantic Ocean using the SEEK assimilation method. The method relies on an ensemble of 24 members. The results show that the models' chlorophyll, which is the variable that is assimilated is improved after assimilation, however not in all regions. In some regions the model variability does not cover the observations and there the assimilation does not improve the chlorophyll. The model results also show that the non-observed variables (nutrients) are not necessarily updated to a better state and in some regions it increases in the upper 100 meters. Finally, they propose a method for alleviating this problem by only applying assimilation to the model fluctuations, this method is demonstrated for one month only.

[Figure]

Overall, I find the paper well written with very interesting results that contribute to the development of data assimilation methods for biogeochemical models and is therefore relevant for OS. However, there are a few things that are unclear, so I propose some minor revisions to this manuscript before it is accepted for publication.

Main comments:

1) Method of generating the ensemble: here the paper simply refers to two publications and refer the reader to those. The method of generating the ensemble is very important and I think the reader deserves a short description of how this was done.

2) Update the paper structure: The method of only assimilating them model fluctuations around the climatology, should be introduced in the method section and the results presented in the result section. Then reserve the discussion section for discussion of the results. I am also confused by the sentence that starts with "For the sake of.. " on line 416, so please clarify what you mean by the climatology in this case. Also specify which period is run. What happens to chlorophyll in this case, is the spread of the profiles increases or does it just appear that way in the figure 9?

3) Trying to understand the results in context of the physical model performance: It would be useful to have some information on the physical models' performance, I am thinking especially of the representation of the extent of the subtropical gyre, since that seems to be a problem area.

Other:

In the title and abstract 'ocean' is spelled with a capital O.

Abstract ". . . are assimilated daily into. . .!

Line 40 and onwards: Please explain the statement: "However, none of the latter studies explicitly incorporates the uncertainties in the ocean biogeochemistry introduced by stochastic approaches." For example Ciavatta et al 2011, generates an ensemble by perturbing the background light attenuation, that would also be considered a stochastic

approach, no? Do you mean that in this case the perturbations are done on the model parameters and not on the forcing?

Line 42: " . . .to what extent. . ."

Line 45: "To that end.."

Line 65: I would characterize 1/4 degree resolution as eddie permitting rather than eddie resolving.

Line 85: the model description mentions iron input from rivers, are any other nutrients supplied from rivers?

Line 106: Do you mean the "biogeochemical system"?

Line 108: Specify which key biogeochemical parameters were perturbed?

Line 115: It should be "cost efficient"

Line 118: "We observed that the. . ."

Line 123 exchange "one-day" with "daily composits"

Line 130 "commented on below. . ."

Line 136 delete "biomass" at the end of the sentence.

Line 215: suggest "minimize" instead on "diminish"

Line 224-225: suggest: "However, there is a too strong gradient between the oligotrophic conditions of the North Atlantic subtropical gyre and temperate waters to the north."

For the discussion: What could be done to reduce the 'stripes' left by the satellite swaths on the DA analysis?

The very northward region would be ice-covered during part of the season, is that included in the model?

Please provide the name of the Longhurst provinces by names in addition to numbers in the figures, there is room for that and it will make the reading of the paper easier.

Figure 6 and 9: it is very difficult to read the black text on the dark blue background, try white text?

Figure 6: In province 6, the deeper nutrients also quite far off from the climatology also in the free run, why is that? Does it persist far down into the deeper layers?

Figure 7: Could you add a third column where you show the isolines of climatological nitrate?

Line 393: ". . ..(1) it significantly reduces . . ."

———————————————————

---

## Author Comment (AC1) · 28 May 2020

We thank the anonymous reviewer of the manuscript for its careful revision and thoughtful comments and suggestions. We have considered all suggestions and responded below to each individual comment. We hope that we have been able to solve the gaps and answer other queries in our revised text.

Reviewer 1: This study presents an ensemble data assimilation scheme for ocean colour, based on stochastic parameterisations and the SEEK filter. A 24-member ensemble is run for one year with and without assimilation, and assessment made of the ensemble spread and fit to observations. The assimilation generally improves both of

these, as long as the prior ensemble spread is sufficient. If not, then the assimilation can degrade unobserved variables. This impact was reduced in a short experiment where the assimilation was applied only to anomalies from model climatology. The paper is interesting and well written, and the assessment clear and balanced. A few things need expanding on or clarifying, as detailed below, but if those are addressed then I recommend publication in Ocean Science.

Major comments: Given that the paper is taking a previously used deterministic assimilation scheme (e.g.Fontana et al., 2013) and turning it into an ensemble scheme, I was surprised that no comparison was made to a static implementation of the SEEK filter. I appreciate that the main focus of the paper is to study the ensemble aspects, and that ensembles give probabilistic information that is more widely useful, but given the 24-fold increase in computational cost, it would be useful to see how the ensemble median compares to a deterministic assimilation run. A deterministic run is mentioned in the text (lines 98-99) but not presented, so hopefully this would not involve too much extra effort. It could either be an extra sub-section of the results, or incorporated into some of the existing figures.

We agree with the reviewer that ensemble data assimilation is computationally much more expensive than the static implementation of the SEEK filter previously used for instance in Fontana et al. (2013), and that this increase in the cost must be compensated by substantial benefits. However, it is important to remark that these two assimilation systems do not exactly solve the same problem. One is only providing one estimated trajectory for the state of the system, while the other is providing a probability distribution. This is already an important benefit because it provides information about uncertainties to the users, and because it allows an objective validation of the system using probabilistic scores (like rank histograms), as it is done in this paper. Second, in this kind of system, the explicit simulation of model uncertainties is necessary to produce a description of uncertainties that is consistent with observations, even in the free simulation, as was shown in Garnier et al. (2016). In a deterministic system, it is

thus very difficult to provide forecast error covariance matrices that are consistent with the real error. Thus, even if the estimated trajectory is not too far from the observations (as in Fontana et al., 2013), this is still a problematic limitation of the assimilation system. For these reasons, we did not try to perform data assimilation experiments with the deterministic assimilation scheme anymore. The deterministic run mentioned in the text in lines 98-99 is a free run, not an assimilation run, and this deterministic free run was compared to an ensemble free run in Garnier et al. (2016), showing that the description of model uncertainties was very important here. No deterministic assimilation run is thus available to make the comparison required by the reviewer. However, the following text has been introduced in line 126 of the paper to better explain this point: "Though using a probabilistic approach is more resource-costing, it produces a probability distribution that allows for an objective validation with observations using probabilistic scores, unlike a deterministic assimilation system that provides only one estimated trajectory. As another advantage, the explicit simulation of model uncertainties in the ensemble approach is necessary to produce a description of uncertainties that is consistent with observations.".

The last paragraph of Section 2.3 briefly states that ensemble sizes of 12, 24 and 60 members were compared, and 24 able to give similar results to 60. The issue of ensemble size is an important one that will be of wider interest, so I think this assessment should be presented in the paper.

Following the suggestion of the reviewer, we have extended the comments on the assessment used to choose a 24 member ensemble as the most appropriate ensemble size for our final system. Next lines replace the text previously dedicated to explaining the sensitivity experiment performed to choose the size of the ensemble: "More explicitly, 1-month assimilation experiments were performed by reducing the ensemble size from the original 60 members to 12 and 24 members. We first compared each of them with the original experiment, and observed surface chlorophyll differences below 0.5 mg Chl m-3 for most regions between the 24 and the 60 member ensembles. A

СЗ

comparison against the observations used for the assimilation process was also assessed. Both reduced ensemble simulations were able to reproduce the main patterns of surface chlorophyll displayed by satellite observations. However, global probabilistic metrics showed that only the 24 member ensemble experiment conserves the same level of statistical consistency as the original ensemble, while reducing computational costs of the forecast step by up to 60%. The probability distribution of the 12 member ensemble showed an underdispersed distribution, while the 24 member ensemble showed the ensemble spread covers a major part of the observations. Therefore, a total of 24 trajectories of the inherited stochastic simulation developed by Garnier et al. (2016) are used here as the prior PDF for the assimilation problem.".

Section 4.3 is very interesting but also brief. It's fine for it to just be a one-month experiment, but it would be useful to expand on both the methodology (e.g. is the seasonal cycle considered in calculating the climatology?) and the assessment (what's the general impact on chlorophyll skill?).

We agree that this section was too short to be clear. To answer this request, this section has been rewritten to provide a more detailed explanation of the method, and to enhance the interpretation of the results.

Minor comments: Figures: Many use a rainbow colour scheme, which is increasingly discouraged (e.g.Hawkins et al., 2015; https://doi.org/10.1038/519291d). There is no "best" colour scheme I can recommend, but it is worth considering if there is a more appropriate colour scheme for these plots.

Following the recommendations of the reviewer, figures 1, 2, and 7 have been replotted using a new colormap.

Figure 1: It would be best to mask out areas which are not in the model domain (e.g. the Pacific and eastern Mediterranean).

Though we agree these areas would be better masked out, to mask them requires

interpolating satellite data into the model mask file or building a specific mask for it. We think the goal of the figure is just to illustrate the region of interest, and thus we consider it is not essential to make this process.

Line 65: "eddy-resolving" should be "eddy-permitting". Line 173: "NOOA" should be "NOAA".

These two suggestions are amended in the new version.

Line 67: "ERA-INTERIM atmospheric fields (Brodeau et al., 2010)." The reference refers to ERA40, not ERA-Interim.

The reference to ERA-Interim has been corrected. Simmons, (2006), and Dee et al., (2011), are cited in the new version.

Lines 98-99: "a deterministic simulation...for a period of six years" – this doesn't seem to be presented?

In order to clarify this statement, this sentence has been included at the end of the paragraph: "This simulation is used to build a probabilistic configuration upon which a data assimilation system is performed."

Lines 107-108: Please provide a little more detail on the perturbations, so the casual reader doesn't need to read the references.

Following the suggestion of the reviewer we have provided more details on the perturbations on the text by adding: "...whose uncertainties may have a direct impact on the estimation of primary production. Specifically, the parameters perturbed are the phytoplankton growth rate at 0°C, the initial P-I slope for both nanophytoplankton and diatoms, the phytoplankton temperature sensitive of growth, the zooplankton temperature sensitive of grazing and the growth dependency to the day length for both nanophytoplankton and diatoms. For the perturbations, the starting point is a first-order autoregressive process setting up with a standard deviation of 0.3 and a decorrelation time scale of 1 month, at which a random noise is drawn at each grid point and at

each time step. After spatial filtering, Gaussian noises are transformed in Lognormal noises to guarantee positivity. Stochastic perturbations are then introduced by multiplying by these Lognormal noises. To preserve vertical consistency, all perturbations are set identical for the whole water column. In addition, as the effects of unresolved scales will have an impact on the large scale biogeochemical representation, we create a perturbation that simulates the unresolved fluctuation of the concentration of each parameter within every model grid box."

Line 163: Worth clarifying that only SeaWiFS, MODIS and MERIS are used for 2005.

This is amended in the text by : "Data from SeaWiFS, MODIS and MERIS sensors are used for year 2005".

Line 170: Remove "completely".

It has been removed.

Line 232: "satellite swaths leave imprints of their trajectory". It would be good to discuss why this might be happening in the Discussions section. Is it due to the 1 degree localisation radius? Does it imply that the increments are not being retained by the model?

As the reviewer points out, this is due to the small localization radius that is used. This radius needs to be small because the horizontal correlation length scale of the forecast uncertainties in the chlorophyll field is also small. Because of this local behaviour of the system, the impact of a given observation on the observational update must remain local as well, and it is difficult to avoid seeing the imprints of the border between the observed and non-observed regions on the updated fields. The fact that this imprint can still be seen in the forecast actually means that the increment is well retained by the model, and that the model keeps it local (over a few days) consistently with what is said above. However, these imprints should progressively disappear with time as more and more observations are assimilated, so that the error in the system and thus

the magnitude of innovation decreases. In our experiment, this does not happen everywhere because the time lag between observations is quite large with respect to the typical time scale of the system. The model error is also substantial, so that innovation does not become small enough to avoid producing quite large increments with a visible imprint of the borders of the observed area. This behaviour of the system is now better explained in the paper: "These imprints are caused by using a small localization radius. This radius needs to be small due to the small correlation length scale of forecast uncertainties in the chlorophyll field. Thus, the impact of a given observation on the update remains local. They should disappear over time as the magnitude of the innovation decreases. In this experiment, however, the time lag between observations is quite large with respect (5 to 7 days) to the typical time scale of the system."

Lines 254-255: "preserves its reliability...showing a better reliability." I understand what's meant, but it's maybe worth rephrasing these two sentences to be clear about how the reliability has/has not changed.

To clarify, these two sentences have been rephrased as "When observations are assimilated, the distribution of ranks flattens with respect to the shape of the histogram of the non-assimilated experiment."

Line 293: "the metric tends to zero". Perhaps I misunderstand what's meant, but it looks to me like this is a seasonal feature, and the resolution is starting to increase again the following spring, rather than it tending to zero and staying near zero.

We have wrongly used the term "tends to zero" in this sentence. We meant that CRPS is close to zero (note values are 10-7) in both the non-assimilated and the assimilated simulation. We have changed this sentence appropriately: "...the metric is close to zero...". As the reviewer correctly noted, the metric follows a seasonal variability that is now commented in the text as: "A marked seasonality is observed in the CRPS time series. During summer, the resolution of both simulations increases until the end of the season when it returns back to lower values."

Figures 6 and 9: In the labels, the black text on dark blue for "Satellite product" is very hard to read, and "WOA2008" is grey in the label but black on the plot. I think it worth altering how the labels are plotted for clarity. Also, what's the reason for the dotted grey line at 50m depth?

Labels have been changed in the figures. The dotted grey line at 50 m depth was a typo. We have amended this as well in the new version of the manuscript.

Figure 7: The x-axes should be labelled with "N" rather than "W". What is the reason for the bold dotted lines at 150m and 200m?

X-axes is now correctly labeled in the new figure. Grid lines and tick label sizes were highlighted for aesthetic reasons. These highlights are removed in the new version.

Line 375: "An inferior boundary that may cause this overestimation." I think this needs expanding on.

The message of the sentence is that when and where concentrations are small, the perturbations can hardly make them decrease thus only producing overestimation of chlorophyll. As this is not essential information, we have decided to remove it from the manuscript.

Line 404: "the ensemble is not stochastic enough". I think a more accurate phrasing would be "the ensemble has insufficient spread" or something similar.

As the reviewer correctly proposed, this has been changed appropriately in the new version by "the ensemble has insufficient spread in provinces...".

Line 409: "a more homogeneous ensemble". Again, I think this needs rephrasing. The histogram is more homogeneous, which means the ensemble has more appropriate spread, rather than being homogeneous itself.

As proposed, this has been rephrased to a more convenient: "As a result, the subsequent daily forecast is based on an ensemble that has more appropriate spread thus improving its general performance.".

Lines 450-454: Is the assimilation just having a weaker impact, or is it having a better impact due to the reduction of model bias in the assimilation?

We think this question may have been solved with the expansion of section 4.3.

Line 462: "unusable". Arguably, but I would suggest "of limited use" or "insufficient".

"Insufficient" is used to substitute the "unusable" term.

---

## Author Comment (AC2) · 28 May 2020

We thank the anonymous reviewer of the manuscript for its careful revision and thoughtful comments and suggestions. We have considered all suggestions and responded below to each individual comment. We hope that we have been able to solve the gaps and answer other queries in our revised text.

Reviewer 2: This paper describes the assimilation of chlorophyll into a model of the North Atlantic Ocean using the SEEK assimilation method. The method relies on an ensemble of 24 members. The results show that the models' chlorophyll, which is the variable that is assimilated is improved after assimilation, however not in all regions.

[Figure]

In some regions the model variability does not cover the observations and there the assimilation does not improve the chlorophyll. The model results also show that the non-observed variables (nutrients) are not necessarily updated to a better state and in some regions it increases in the upper 100 meters. Finally, they propose a method for alleviating this problem by only applying assimilation to the model fluctuations, this method is demonstrated for one month only.

Overall, I find the paper well written with very interesting results that contribute to the development of data assimilation methods for biogeochemical models and is therefore relevant for OS. However, there are a few things that are unclear, so I propose some minor revisions to this manuscript before it is accepted for publication.

Main comments: 1) Method of generating the ensemble: here the paper simply refers to two publications and refer the reader to those. The method of generating the ensemble is very important and I think the reader deserves a short description of how this was done.

As suggested by the reviewer, a simplified explanation of the methodology employed in Garnier et al. (2016) to generate the ensemble is included in the text: "...whose uncertainties may have a direct impact on the estimation of primary production. Specifically, the parameters perturbed are the phytoplankton growth rate at $0^\circ$ C, the initial P-I slope for both nanophytoplankton and diatoms, the phytoplankton temperature sensitive of growth, the zooplankton temperature sensitive of grazing and the growth dependency to the day length for both nanophytoplankton and diatoms. For the perturbations, the starting point is a first-order autoregressive process setting up with a standard deviation of 0.3 and a decorrelation time scale of 1 month, at which a random noise is drawn at each grid point and at each time step. After spatial filtering, Gaussian noises are transformed in Lognormal noises to guarantee positivity. Stochastic perturbations are then introduced by multiplying by these Lognormal noises. To preserve vertical consistency, all perturbations are set identical for the whole water column. In addition, as the effects of unresolved scales will have an impact on the large scale biogeochemical

representation, we create a perturbation that simulates the unresolved fluctuation of the concentration of each parameter within every model grid box.".

2) Update the paper structure: The method of only assimilating the model fluctuations around the climatology, should be introduced in the method section and the results presented in the result section. Then reserve the discussion section for discussion of the results. I am also confused by the sentence that starts with "For the sake of.. " on line 416, so please clarify what you mean by the climatology in this case. Also specify which period is run. What happens to chlorophyll in this case, is the spread of the profiles increases or does it just appear that way in the figure 9?

We considered the structure proposed by the reviewer when first starting to write the manuscript. However, we finally decided to present first the results of the main two simulations (the free run and the assimilated) and discussed them, and then explain the sensitivity experiment developed in order to cope with the inconsistencies found in the system. We think this structure is the most appropriate as we first present the strengths and weaknesses of the system, and then try to solve them by only assimilating model fluctuations. Section 4.3. has been rewritten to provide a more detailed explanation of the method, and to enhance the interpretation of the results. We expect these issues are clearer after modifications.

3) Trying to understand the results in context of the physical model performance: It would be useful to have some information on the physical models' performance, I am thinking especially of the representation of the extent of the subtropical gyre, since that seems to be a problem area.

We agree with the reviewer in that physical model performance is of utmost importance to understand some general flaws of our system. The eddy-permitting physical model used in the present study was first documented in Barnier et al. (2006), in the context of numerical schemes tested in a global, $\frac{1}{4}°$ configuration to reduce the known biases in the representation of western boundary currents and subtropical gyres, such

as in the North Atlantic. In Ourmières et al. (2009), a detailed analysis was made on the mixed layer dynamics at mid-latitudes, which is known to have a decisive impact on primary production. Comparisons were made between a free simulation (as in the present study), experiments with assimilation of physical observations (SST, altimetry) and climatological data (T, S and nutrients), and a seasonal climatology of mixed layer depth. They observe that in March, the free solution exhibits a too deep mixed layer extended over an abnormally large area compared to the climatology, in the Gulf Stream region and its north eastern extension. In April, stratification takes place in the model, in agreement with the climatology, despite a remaining area of large MLD east of Cape Hatteras and an overestimated zone in the north-east Atlantic above 45°N. They also show that the combined assimilation of physical and nutrient data has a positive impact on the phytoplankton patterns by comparison with SeaWiFS ocean colour data. It is obvious that this biased representation of the dynamics has a significant impact on the results of our study, which is dedicated solely to the control of sources of uncertainty in the biological model. Therefore, in the revised manuscript, we refer more explicitly to the MLD analysis of Ourmières et al. (2009).

Other: In the title and abstract 'ocean' is spelled with a capital O.

This has been amended in the new version.

Abstract ". . . are assimilated daily into. . .!

Line 42: " . . .to what extent. . ."

Line 45: "To that end.."

Line 115: It should be "cost efficient"

Line 118: "We observed that the. . ."

Line 123 exchange "one-day" with "daily composites"

Line 130 "commented on below. . ."

Line 136 delete "biomass" at the end of the sentence.

Line 215: suggest "minimize" instead on "diminish"

Line 393: ". . ..(1) it significantly reduces . . ."

Precedent suggestions have been taken into account and changed accordingly in the new version of the manuscript.

Line 40 and onwards: Please explain the statement: "However, none of the latter studies explicitly incorporates the uncertainties in the ocean biogeochemistry introduced by stochastic approaches." For example Ciavatta et al 2011, generates an ensemble by perturbing the background light attenuation, that would also be considered a stochastic approach, no? Do you mean that in this case the perturbations are done on the model parameters and not on the forcing?

In this system, in contrast to similar works, the ensemble has been explicitly developed by introducing perturbation on model parameters. To clarify this point, we have added a short comment on the manuscript: "However, none of the latter studies explicitly incorporates the uncertainties in the ocean biogeochemistry introduced by stochastic approaches on the model formulation.".

Line 65: I would characterize 1/4 degree resolution as eddie permitting rather than eddie resolving.

Following the suggestion of the reviewer, we have changed the term to "eddy permitting".

Line 85: the model description mentions iron input from rivers, are any other nutrients supplied from rivers?

Yes, riverine inputs of other nutrients are also taken into account by PISCES. As a default, river supply of all elements but DIC and alkalinity is taken from GLOBAL-NEWS2 (Mayorga et al., 2010). We consider that mentioning iron inputs is not relevant for the

sake of this manuscript, and it has been removed in this new version.

Line 106: Do you mean the "biogeochemical system"?

Seven biogeochemical parameters were perturbed to include uncertainties arising from the limitation of the simulation to describe the biogeochemical system. We have included explicitly in this version that we refer to the biogeochemical system by "..the simplification of the description of the biogeochemical system to a limited number of state variables and parameters.".

Line 108: Specify which key biogeochemical parameters were perturbed?

The parameters that were perturbed in the system presented in Garnier et al. (2016), and that we use here to build the subsequent assimilation system are presented now in the text as: "...whose uncertainties may have a direct impact on the estimation of primary production. Specifically, the parameters perturbed are the phytoplankton growth rate at $0°$ C, the initial P-I slope for both nanophytoplankton and diatoms, the phytoplankton temperature sensitive of growth, the zooplankton temperature sensitive of grazing and the growth dependency to the day length for both nanophytoplankton and diatoms.".

Line 224-225: suggest: "However, there is a too strong gradient between the oligotrophic conditions of the North Atlantic subtropical gyre and temperate waters to the north."

The sentence has been changed as suggested.

For the discussion: What could be done to reduce the 'stripes' left by the satellite swaths on the DA analysis?

Satellite imprints are due to the small localization radius that is used. This radius needs to be small because the horizontal correlation length scale of the forecast uncertainties in the chlorophyll field is also small. Because of this local behaviour of the system, the impact of a given observation on the observational update must remain local as well,

and it is difficult to avoid seeing the imprints of the border between the observed and non-observed regions on the updated fields. The fact that this imprint can still be seen in the forecast actually means that the increment is well retained by the model, and that the model keeps it local (over a few days) consistently with what is said above. However, these imprints should progressively disappear with time as more and more observations are assimilated, so that the error in the system and thus the magnitude of innovation decreases. In our experiment, this does not happen everywhere because the time lag between observations is quite large with respect to the typical time scale of the system. The model error is also substantial, so that innovation does not become small enough to avoid producing quite large increments with a visible imprint of the borders of the observed area. This behaviour of the system is now better explained in the paper: "These imprints are caused by using a small localization radius. This radius needs to be small due to the small correlation length scale of forecast uncertainties in the chlorophyll field. Thus, the impact of a given observation on the update remains local. They should disappear over time as the magnitude of the innovation decreases. In this experiment, however, the time lag between observations is quite large with respect (5 to 7 days) to the typical time scale of the system."

The very northward region would be ice-covered during part of the season, is that included in the model?

Yes, the physical model code corresponds to NEMO version 3.4. This model benefits from the LIM-2 sea ice parameterization presented in Fichefet and Maqueda (1997), which includes the most relevant thermodynamics (exchanges of heat) and dynamics (exchanges of mass and momentum) sea ice-related processes.

Please provide the name of the Longhurst provinces by names in addition to numbers in the figures, there is room for that and it will make the reading of the paper easier.

Longhurst provinces are now named both in Figure 1 and its label, and into the text. Label of Figure 1 now includes: "Provinces indicated by acronyms NADR (North Atlantic Drift), NATR (North Atlantic tropical gyre), NASTE (Northeast Atlantic subtropical gyre), and NASTW (Northwest Atlantic subtropical gyre) are used throughout the text.". In the text we have included: "The provinces used are NADR (North Atlantic Drift), NATR (North Atlantic tropical gyre), NASTE (Northeast Atlantic subtropical gyre), and NASTW (Northwest Atlantic subtropical gyre).". Then, acronyms substitute province numbers throughout the manuscript.

Figure 6 and 9: it is very difficult to read the black text on the dark blue background, try white text?

Labels have been conveniently modified in the figures.

Figure 6: In province 6, the deeper nutrients also quite far off from the climatology also in the free run, why is that? Does it persist far down into the deeper layers?

The climatology used to compare with model data contains data up to 800 m deep. Through the water column between surface and this depth, nutrients are overestimated consistently in province 6. The shape is reproduced but values are far from climatological data. One possible explanation for this overestimation is that comparisons are made between daily snapshots extracted from our assimilated and non-assimilated simulations, and an observational-based climatology. On the other hand, the deterministic simulation already overestimates chlorophyll in province 6 (NASTW) as observed in Garnier et al., 2016. As commented above, Ourmières et al. (2009) carried out a detailed analysis on the mixed layer dynamics at mid-latitudes. They observed a band with too high concentrations of nitrate in the northern part of this region (35-40° N) that triggers a chlorophyll overshoot in the following months as we observed here.

Figure 7: Could you add a third column where you show the isolines of climatological nitrate?

Figure 7 was intended to show the effects of assimilation by comparing the water column before and after the procedure. The WOA nitrate fields correspond to monthly

climatological data, and offer a coarse grid resolution of 1°. We consider that a section from this data is not appropriately comparable to the ensemble simulations presented here.

---

## Author Response (AR2)

**Reply to the Topic Editor**

We thank the Topic Editor of the manuscript for its careful revision and thoughtful comments and suggestions. We have considered all of their suggestions and responded below (in blue) to each individual comment (in black). We hope that we have been able to solve all technical issues suggested in our revised text.

Comments to the Author:

The authors have adequately addressed the issues/concerns of the reviewers. I only have a few minor technical corrections for the authors to consider.

Minor Suggestions

line 83 "We refer Chl-a to here as the direct sum of these two compartments, and it will be used as a proxy for primary production." change to "Here we ascribe Chl-a as the direct sum of these two compartments, and it will be used as a proxy for primary production."

Line 90 "that on-line coupling means here one-way forcing .." change to "that on-line coupling is a one-way forcing …"

Line 175 "The observation data set…" change to "The observational data set …"

Line 177 " that consist on daily-accumulated …" change to "..that consist of daily-accumulated .."

Line 182 "(Data from SeaWiFS," change to "(data from SeaWiFS,"

Line 376 "…. to simulate using an holistic modelling system…" change to "….to simulate using a holistic modelling system…"

Line 411 "In this regard, it is important to notice that our multivariate analysis scheme allows corrections on five nutrients, …" change to "In this regard, it is important to notice that our multivariate analysis scheme allows corrections of five nutrients, …"

Line 480 "As shown throughout the text, the assimilated system has brought us promising results." change to "As shown, the assimilated system has provided promising results.

Precedent suggestions have been taken into account and modified accordingly in the new version of the manuscript.

Line 511. "….prospects of BIO-Argo floats …" Check, these float are now more commonly referred to as BGC-Argo, https://biogeochemical-argo.org/. Suggest change to . "….prospects of BGC-Argo floats …", here and elsewhere.

As suggested, BIO-Argo is now referred to as BGC-Argo throughout the text following the most commonly used term.